# Elabela: Negative Regulation of Ferroptosis in Trophoblasts via the Ferritinophagy Pathway Implicated in the Pathogenesis of Preeclampsia

**DOI:** 10.3390/cells12010099

**Published:** 2022-12-26

**Authors:** Huan Yang, Xuemei Zhang, Yubin Ding, Hui Xiong, Shaojian Xiang, Yang Wang, Huanhuan Li, Zheng Liu, Jie He, Yuelan Tao, Hongbing Yang, Hongbo Qi

**Affiliations:** 1Department of Obstetrics, Chongqing University Three Gorges Hospital, Chongqing 404100, China; 2Joint International Research Laboratory of Reproduction and Development of the Ministry of Education of China, Chongqing Medical University, Chongqing 400016, China; 3Department of Obstetrics, The First Affiliated Hospital of Chongqing Medical University, Chongqing 400016, China; 4Department of Obstetrics and Gynecology, Women and Children’s Hospital of Chongqing Medical University, Chongqing 401147, China; 5Department of Emergency, Chongqing University Three Gorges Hospital, Chongqing 404100, China

**Keywords:** preeclampsia, trophoblast, Elabela, ferroptosis, ferritinophagy

## Abstract

Preeclampsia is a leading contributor to increased maternal morbidity and mortality in the perinatal period. Increasing evidence demonstrates that ferroptosis is an essential mechanism for the pathogenesis of preeclampsia. Elabela is a novel small-molecule polypeptide, mainly expressed in embryonic and transplacental tissues, with an ability to promote cell proliferation and invasion. However, its specific regulatory mechanism in preeclampsia has not been completely elucidated. In this study, we first reveal an increased grade of ferroptosis accompanied by a downregulation of the expression of Elabela in preeclampsia placentas. We then confirm the presence of a ferroptosis phenotype in the placenta of the mouse PE-like model, and Elabela can reduce ferroptosis in the placenta and improve adverse pregnancy outcomes. Furthermore, we demonstrate that targeting Elabela alleviates the cellular dysfunction mediated by Erastin promoting increased lipid peroxidation in vitro. Subsequent mechanistic studies suggest that Elabela increases FTH1 levels by inhibiting the ferritinophagy pathway, and consequently chelates the intracellular labile iron pool and eventually arrests ferroptosis. In conclusion, Elabela deficiency exacerbates ferroptosis in the placenta, which is among the potential mechanisms in the pathogenesis of preeclampsia. Targeting the Elabela–ferritinophagy–ferroptosis signaling axis provides a new therapeutic intervention strategy to alleviate preeclampsia.

## 1. Introduction

The overall incidence of preeclampsia (PE) is approximately 2–5%. The disease is characterized by proteinuria and hypertension, which seriously threaten maternal and fetal safety [1,2]. The pathogenesis of PE is complex. Inadequate trophoblast invasion and ineffective remodeling of the uterine spiral arteries in the presence of abnormal trophoblast function are the prevailing theories for the development of PE [3,4]. Therefore, trophoblast malfunction deserves attention when investigating the etiology of PE.

Ferroptosis is a newly discovered form of programmed cell death, mainly caused by iron-dependent lipid peroxidation leading to lipid membrane damage, which significantly differs from apoptosis, necrosis, and autophagy in terms of morphology, biochemistry, and genetics [5]. This specific mode of lipotoxic cell death is associated with many human diseases, including cancer, ischemia–reperfusion heart injury, brain injury, acute kidney injury, and asthma [6,7,8,9,10]. Ferroptosis is widely detected in the villi and extravillous trophoblasts of the placenta of pregnancies with PE or Fetal growth restriction (FGR), suggesting moderate ferroptosis in trophoblasts is essential for a healthy pregnancy [11,12,13,14,15], although how ferroptosis of trophoblasts leads to pathological progression of the placenta and its pathological mechanisms are not yet evident in PE. However, ferroptosis can lead to excessive trophoblast death, disrupt normal extravillous cytotrophoblast renewal, and prevent the normal recruitment of syncytiotrophoblasts, which is beyond doubt. There is evidence that factors such as oxidative stress and hypoxic reoxygenation injury, which plays a role in the development of PE, can also induce iron death in in vitro-cultured villous trophoblast cells [11,16]. Inhibition of ferroptosis can reduce the lipid peroxidation and oxidative stress of trophoblasts, and increase the proliferation and invasion ability of cells [13]. Reversal of ferroptosis can effectively decrease placental lipotoxic damage and alleviate the symptoms associated with PE in vivo [11,14,17]. These results indicate that ferroptosis is a potentially critical mechanism responsible for placental dysfunction. Therefore, negative regulation of ferroptosis may be beneficial in alleviating or treating PE due to placental dysfunction.

Elabela is an endogenous active peptide, first discovered in zebrafish embryos by Chng et al. in 2013 [18], and expressed in vertebrate embryos, placenta, kidney, blood vessels, heart, and various tumor tissues [19,20,21,22]. Elabela can interact with angiotensin II (AngII) type 1 receptor-related protein (APJ) to regulate corresponding physiological functions in organisms. Elabela can maintain the growth and self-renewal of embryonic stem cells by activating the phosphatidylinositol 3-kinase/serine-threonine kinase (PI3K/AKT) pathway to promote protein translation, cell cycle progression and inhibit cell death [23]. The placenta of Elabela-deficient mice is hypoplastic, characterized by a thinner placental labyrinth level, smaller placental volume, etc., and exhibits symptoms similar to preeclampsia. Exogenous Elabela supplementation reduced blood pressure and improved proteinuria symptoms in mice [24]. 

Although several studies had confirmed that Elabela could reverse disease progression by reducing the extent of ferroptosis, no data were available on PE [25,26,27]. Therefore, in this study, we intended to explore the link between Elabela and ferroptosis during PE and the mechanisms that mediate it.

## 2. Materials and Methods

### 2.1. Study Participants and Sample Collection

This study was approved by the Ethics Committee of Chongqing Medical University (No: 2020-790) following the principles set out in the Declaration of Helsinki. Written informed consent was obtained from all patients. The inclusion criteria for cases were referenced from the 2019 American College of Obstetricians and Gynecologists (ACOG) guidelines [28]. We recruited pregnant women with severe PE (*n* = 20) and healthy pregnant women (*n* = 20). Their detailed clinical characteristics are shown in Appendix A.

### 2.2. Animal

The 8–12-wk-old CD-1 female mice weighing 25–35 g from the Chongqing Byrness Weil Biotechnology Co., Ltd. (Chongqing, China) were mated with age-matched male mice. All mice are raised in the animal care facility of Chongqing Medical University in a temperature-controlled room (23 °C) with a light–dark cycle of 12:12 h. Once the vaginal peg was detected, the day was identified as E0.5. Pregnant mice were randomly divided into control, L-NAME or L-NAME + ELA groups. From E8.5 to E18.5, the L-NAME and L-NAME + ELA groups were administered 100 mg/kg/day L-NAME (Cat. N5751; Sigma-Aldrich, St. Louis, MO, USA) by intragastrical [29]. In addition, the L-NAME+ELA group was treated with intraperitoneal injection with ELA-32 1 mg/kg/day (Cat. 1680205-79-1; Tocris Bioscience, Bristol, UK) [27]. From E6.5 to E18.5, systolic blood pressure (SBP) was monitored daily by tail-cuff photoplethysmography BP-2000 Series II (Visitech Systems, Apex, NC, USA). On E18.5, mice were euthanized for further experiments. All experimental procedures involving animals were approved by the Ethics Committee of Chongqing Medical University.

### 2.3. Cell Culture

The immortalized HTR8-S/Vneo trophoblast cell line was obtained from American Type Culture Collection (ATCC, Manassas, VA, USA) and grown in Roswell Park Memorial Institute (RPMI) 1640 (Cat.11875093; Gibco, Grand Island, NY, USA) supplemented with 10% FBS (Cat. ST30-2602; PAN-Biotech, Adenbach, Germany) and 1% penicillin-streptomycin (Cat. C0222; Beyotime, Shanghai, China). All the cells were cultured at 37 °C in 5% CO_2_ humidified air. ELA32 (Cat. 1680205-79-1; Tocris Bioscience, Bristol, UK), H2O2 (Cat. 323381, Sigma-Aldrich, MO, USA), and LPS (Cat. L5293; Sigma-Aldrich, MO, USA) were dissolved in double-distilled water, while Erastin (Cat. S7242; Selleck, Houston, TX, USA), RSL3(Cat. S8155; Selleck, Houston, TX, USA), Z-DEVD-FMK (Cat. S7312; Selleck, Houston, TX, USA), Necrostatin-1(Cat. S8037; Selleck, Houston, TX, USA), Ferrostatin-1 (Cat. S7243; Selleck, Houston, TX, USA), and Deferoxamine-1 (Cat. S5742, Selleck, Houston, TX, USA) were dissolved in dimethyl sulfoxide (DMSO) (Cat. D1435; Sigma-Aldrich, MO, USA).

### 2.4. Transfection

Small interfering (si)-ATG5, si-NCOA4, and a negative control siRNA (si-NC) were synthesized by Tsingke Biotechnology (Beijing, China). The FTH1-EGFP plasmid, the ATG5 overexpression plasmid, the NCOA4 overexpression plasmid, and a negative control overexpression plasmid (OE-NC) were synthesized by Hanbio Biotechnology (Shanghai, China). HTR8/Svneo cells at 60–70% confluency was transfected with 100 nM siRNA or 1 mg of plasmids in the presence of Lipofectamine 2000 (Thermo Fisher Scientific, Waltham, MA, USA) in six-well plates according to the manufacturer’s instructions. The siRNA sequences used in this study are listed in Appendix A. 

### 2.5. Immunohistochemistry

Dehydrated and paraffin-embedded placental tissues were cut into 4 m thick slices after being rinsed in PBS, fixed overnight at room temperature in 4% paraformaldehyde, and fixed again the following day. Deparaffinization and rehydration of tissue slices were done using a graduated alcohol series. Antigen repair was achieved by microwave treatment of the sections in 10 mM sodium citrate (pH 6.0) for 15 min. The slices were then treated for 15 min at room temperature with 3% H_2_O_2_ to inhibit endogenous peroxidase activity. Next, the slices were incubated with rabbit monoclonal antibody against ELA (1:100; Cat. H-007-19; Phoenix Pharmaceuticals, Burlingame, CA, USA), rabbit monoclonal antibody against 4-HNE (1:200; Cat. BS-6313R, Bioss Antibodies, Woburn, MA, USA), mouse monoclonal antibody against cytokeratin 7 (CK7) (1:200; Cat. 17513-1-AP; Proteintech, Wuhan, China) and mouse monoclonal antibody against human leukocyte antigen G (HLA)—G) (1:100; Cat. MEM-G/1; Proteintech, Wuhan, China) overnight at 4 °C. Afterward, RT peroxidase was treated with horseradish coupled secondary antibody for 30 min. Finally, the immune complexes were visualized using diaminobenzidine. These images were taken on an EVOS microscope (Life Technologies, Carlsbad, CA, USA). The intensity of positive staining in each sample was measured using ImageJ1.50i software. Briefly, 3 random fields from 3 different experiments were quantified per sample [30].

### 2.6. Perls’ Prussian Blue Staining

Referring to previous reports, Prussian staining was performed to determine iron accumulation in placental tissue [31]. Briefly, sections were incubated in Perls’ solution for 30 min after deparaffinization, followed by washing 3 times in PBS. A methanol solution containing 0.3% H_2_O_2_ was then used to suppress the endogenous peroxidase activity for 15 min, followed by washing 3 times in PBS. Signals were developed by incubation in 3,3-diaminobenzidine (DAB) for 15 min and counterstained with hematoxylin. Staining was captured by EVOS microscopy (Life Technologies, Carlsbad, CA, USA). The brown signal was defined as a positive signal and 3 random fields of view for each sample from 3 independent experiments were quantified using ImageJ 1.50i software. In brief, the area of positive signal was quantified for each sample.

### 2.7. H&E Staining

Placentas and kidneys were sectioned into 3 mm-thick pieces after being fixed in 4% paraformaldehyde. Sections were deparaffinized, rehydrated, stained for 5 min with hematoxylin, then for 2 min with eosin. Using an EVOS microscope, pictures were captured (Life Technologies, Carlsbad, CA, USA).

### 2.8. Assay for GSH, MDA and Iron

Kits were used to measure the levels of GSH (Cat. A006-1-1; Nanjing Jiancheng Bioengineering Institute, Nanjing, China), MDA (Cat. S0131S; Beyotime Biotechnology, Shanghai, China), and iron (Cat. E1042-100; Applygen Technology, Beijing, China) in placental tissues and cells according to the manufacturer’s instructions. Briefly, after the lysis of tissue samples or cells, the supernatant was collected after centrifugation at 10,000–12,000 g for 10 min at 4 °C. The content was measured on a colorimetric microplate reader (Thermo Scientific, Multiskan, FC, USA). Protein concentrations were measured using the kit according to the manufacturer’s instructions (Cat. P0010; Beyotime Biotechnology, Shanghai, China). Then, the GSH, MDA and iron values were homogenized by protein concentration.

### 2.9. Elabela Elisa

Plasma Elabela expression levels in pregnancy were measured according to the Elisa kit manufacturer’s instructions (Cat. ml059914; Enzyme-linked Biotechnology, Shanghai, China) and levels were measured on a colorimetric enzyme marker (Thermo Scientific, Multiskan, FC, MA, USA).

### 2.10. RNA Extraction and RT-qPCR

According to the manufacturer’s instructions, total RNA was isolated from tissues using TRIzol reagent (Cat. 15596018; Invitrogen, San Diego, CA, USA). Then, the mRNA was reversed into cDNAs according to the manufacturer’s protocols (Cat. 4897030001; Roche, Basel, Switzerland). The primers for Elabela and β-actin are listed in Appendix A. The expression of the transcripts was normalized to the levels of β-actin. SYBR Green dye (Cat. 4913914001; Roche, Basel, Switzerland) was then used in real-time PCR with an Applied Biosystems PCR cycler (Bio-Rad, Hercules, CA, USA). All samples were run in triplicates.

### 2.11. Cell Counting Kit-8 Assay

Cell viability was measured using a CCK8 kit (Cat. HY-K0301; MedChemExpress, Monmouth Junction, NJ, USA) following the manufacturer’s instructions. In a 96-well plate, 5000 cells in a volume of 100 μL each well were grown in a medium containing 10% FBS. Following treatment, 10 μL of CCK-8 reagent was added to each well, and after 4 h of incubation, the samples were analyzed at 450 nm using a microplate reader (Thermo Fisher Scientific, MA, USA).

### 2.12. Determination of Reactive Oxygen Species

Cells were treated as indicated, and then incubated with the 2,7-dichlorodihydrofluorescein diacetate (DCFH-DA) fluorescent probe (Cat. S0033S; Beyotime, Shanghai, China) in a serum-free medium following the manufacturer’s protocols. Cells were incubated for 2 h in the dark, washed twice to remove the probe that did not enter the cells, and then the dishes were covered with a small amount of serum-free medium. ROS fluorescence was captured by EVOS microscopy (Life Technologies, Carlsbad, CA, USA). Cells were treated under the same conditions, digested and collected, and ROS was analyzed using a flow cytometer (CytoFlex, Bckmancoulter, CA, USA). A minimum of 10,000 cells was analyzed per condition.

### 2.13. Assessment of Lipid Peroxidation

Cells were treated as indicated, and then incubated with 50 μM C11-BODIPY™ 581/591 probe (Cat. L267, Dojindo, Japan) away from light for 1 h. Cells were washed twice with PBS to remove the excess probe, and then a little serum-free medium was added to cover the cells in the culture dish. ROS fluorescence was captured by EVOS microscopy (Life Technologies, Carlsbad, CA, USA). Three random fields per sample from 3 different experiments were quantified using ImageJ 1.50i software.

### 2.14. Cellular Labile Iron Detection

FerroOrange probes (Cat.F374; Dojindo, Japan) were used to measure cellular ferrous iron levels. Briefly, cells were seeded in 24-well plates, and treated as indicated; 1 mM FerroOrange was added into the dish and incubated for 30 min away from light. Then, cells were rinsed twice with PBS to remove the excess probe, and ferrous iron fluorescence was captured through an EVOS microscope (Life Technologies, Carlsbad, CA, USA). Three random fields of per sample from 3 different experiments were quantified using ImageJ 1.50i software (NIH, Bethesda, MD, USA).

### 2.15. Cellular Lysosomes Detection

Cells were seeded in 24-well plates, and treated as indicated; 50 nM Lyso-Tracker Green (Cat.C1047S, Beyotime Biotechnology, Shanghai, China) was added to the cells and incubated for 30 min protected from light. and the cells were rinsed twice with PBS to remove the excess probe. Lysosome’s fluorescence was captured by EVOS microscopy (Life Technologies, Carlsbad, CA, USA). Three random fields per sample from 3 different experiments were quantified using ImageJ 1.50i software.

### 2.16. Transmission Electron Microscopy

Cells were fixed in 3% glutaraldehyde followed by 1% osmium tetroxide, gradually dehydrated in a gradient of ethylene oxide, embedded in Ep812 and cured at 60 °C for 24 h. Before transmission electron microscopy, ultrathin slices (50 nm) were put on a 200-mesh copper grid and double stained with uranyl acetate and lead citrate (JEM-1400FLASH, Tokyo, Japan).

### 2.17. Autophagy Flux Analysis

HTR8/SVneo cells at 60-70% confluency was infected with Ad-mCherry-GFP-LC3B (Cat. C3011, Beyotime Biotechnology, Shanghai, China) in 24-well plate at a multiplicity of infection of 20 for 24 h according to the manufacturer’s instructions. Following indicated treatment, autophagy flux was observed under EVOS microscopy (Life Technologies, Carlsbad, CA, USA). Autophagy flux was evaluated by calculating the number of yellow and red dots using ImageJ 1.50i software.

### 2.18. Western Blotting

The method of Western blotting was performed referring to our previous research protocol [32]. Primary antibodies against FPN1 (1:500; Cat. A14885) were purchased from ABclonal (Wuhan, China). FTH1 (1:1000; Cat. ab75973), GPX4(1:1000; Cat. ab125066), Beclin1 (1:1000; Cat. ab210498), ATG5 (1:2000; Cat. ab108327), LC3B (1:2000; Cat. ab192890), P62 (1:10000; Cat.109012) and NCOA4 (1:5000; Cat. ab86707) were purchased from Abcam (Cambridge, UK). TFR (1:500; Cat. WL03500) and GFP (1:500; Cat. WL03189) were purchased from Wanleibio (Shengyang, China). β-tubulin (1:5000, Cat. 10068-1-AP), β-actin (1:5000; Cat. 81115-1-RR) and GAPDH (1:10000; Cat.60004-1-Ig) were purchased from Proteintech (Shanghai, China). 

### 2.19. Matrigel Invasion Assay

HTR8/SVneo cells (50,000 cells/well) were resuspended in FBS-free RPMI1640 media and planted onto the upper compartment of a 24-well plate covered with pre-diluted Matrigel (Corning, New York, NY, USA). After 24 h, the upper chamber was fixed with 4% paraformaldehyde, rinsed with PBS, and dyed with crystalline violet borate. Using an EVOS microscope (Life Technologies, Carlsbad, CA, USA), the cleaned upper chamber was imaged and ImageJ1.50i software was used to determine cell counts.

### 2.20. Wound Healing Assay

HTR8/SVneo cells were seeded in 6-well plates and developed to a confluence level of more than 90%. A linear scratch or wound was made across the confluent monolayer using a fine 100 µL pipette tip, and pictures were taken at 0 and 12 h. The wound healing area was measured using ImageJ 1.50i software.

### 2.21. DNA Synthesis Assay

HTR8/SVneo cells were seeded in 96-well plates at a density of 5000 cells/well. After being treated as indicated, 100 ml of medium containing 50 mM EdU was added to each well for 2 h. Cells were then fixed in 4% formaldehyde for 30 min. After being washed, cells were incubated with Click-iTR EdU Kit (Cat. C10310-1; RiboBio, Guangzhou, China) for 30 min at room temperature, nuclei were stained with Hoechst for 30 min and images were taken with an EVOS microscope (Life Technologies, Carlsbad, CA, USA). The number of EdU-positive cells was counted in 3 random fields of view for each sample in 3 independent experiments using ImageJ 1.50i software.

### 2.22. mRNA Sequencing

HTR8/SVneo cells were grown to 60–70% confluency and then exposed to 5 μM Erastin in the absence or presence of 0.01 μM Elebala in RPMI 1640 with 10% FBS for 12 h. Total RNA was extracted using TRIzol reagent (Cat. 15596018; Invitrogen, CA, USA), and the concentration was measured by Agilent 2100 Bioanalyzer (Agilent Technologies 2100 system, Santa Clara, CA, USA). A total of 1 ug of RNA per sample was used as the input material for the sample preparations. Finally, the index codes were clustered with a NEBNext^®^ Ultra^TM^ Directional RNA Library Prep Kit for Illumina^®^ (NEB, USA). The sequenced library preparations were performed for 150 cycles on the former platform. The differential expression analysis of two samples was performed using the DEGs R package (1.16.1). Significantly differential expression was defined based on *p* < 0.01 and |log2(foldchange)| > 1 as the default thresholds. Gene Ontology (GO), Kyoto Encyclopedia of Genes and Genomes (KEGG) and Gene Set Enrichment Analysis (GSEA) were performed using the clusterProfiler R package.

### 2.23. Statistical Analyses

All data were collected using Prism 8.02 software (GraphPad). All data are presented as the mean ± SEM. Data from two groups were statistically compared using the independent Student’s *t*-test, and multiple groups were statistically compared using one -way ANOVA followed by Tukey’s multiple comparisons. For multiple groups with multiple characteristics, two-way ANOVA was used. *p* < 0.05 was considered statistically significant.

## 3. Results

### 3.1. More Severe Ferroptosis Presented in the Placentas of PE

Adverse pregnancy outcomes in PE patients are associated with placental dysfunction [3,4], so whether ferroptosis exists in the placenta of PE pregnancy is worth exploring. The results showed an increase in MDA and iron levels in PE placenta tissue (Figure 1a,b) and a decrease in GSH levels (Figure 1c) compared to the control group. Besides, Perls’ blue staining showed that the number of iron-positive cells was significantly increased in the placenta of PE pregnancies (Figure 1d). Consistently, we found that 4-HNE was expressed in both placental cytotrophoblasts (CTBs), syncytiotrophoblasts (STBs), and interstitial extravillous trophoblasts (iEVTs), and that PE placentas showed higher levels of expression than normal pregnant placentas (Figure 1e). These results suggest that ferroptosis may be engaged in the pathogenesis of PE.

### 3.2. Low Levels of Elabela in the Placenta and Circulation of the Pregnancy with Preeclampsia

First, IHC staining suggested that Elabela was abundantly presented in the CTBs and STBs, and weakly expressed in the iEVTs (Figure 2a,b). Importantly, IHC staining also suggested that the expression of Elabela was compromised in CTBs, STBs and iEVTs collected from PE pregnancy’s placentas. Similarly, ELISA results showed that circulating levels of Elabela were also lower in pregnancies with PE (Figure 2c). Next, the transcript level of Elabela in the placenta was detected by PCR, and the results showed that the transcript level of Elabela in preeclampsia was also significantly lower than that in the control group (Figure 2d). These results demonstrated a correlation between Elabela and the pathogenesis of PE, possibly due to the downregulation of Elabela.

### 3.3. The PE-like and Ferroptosis Phenotypes in Mice Were Relieved by Elabela Administration

We first analyzed the structure and biological characteristics of Elabela through the RSCB-PDB server (https://www.rcsb.org; access on 11 November 2022). We found that Elabela is characterized by a small molecular weight and is highly conserved (Figure 3a). Referring to previous studies, we first used L-NAME administration to construct a mouse model of PE [29,33], and on this basis, Elabela treatment was administered (Figure 3b). The results showed that the mean SBP was significantly higher in the L-NAME-treated mice compared to the control group. In contrast, Elabela significantly reversed the increase in blood pressure (Figure 3c). At E18.5, the L-NAME-treated group showed lower fetal and placental weights and Crown-rump length (CRL) of fetuses, which were partially reversed by Elabela (Figure 3d–f). In addition, Elabela reversed the glomerular constriction resulting from the L-NAMA administration (Figure 3g). Importantly, placental morphological examination showed that Elabela also rescued the reduced placental vagal layer and vagal/connective layer ratio in mice caused by L-NAMA (Figure 3h).

Meanwhile, we found that Perls’ blue staining showed that the number of iron-positive cells was significantly increased in the placenta of the L-NAME groups (Figure 3i), which is consistent with the finding that Elabela also reduced MDA and iron levels and increased GSH levels in the placenta of mice (Appendix A). In addition, IHC staining revealed that Elabela somewhat reduced L-NAME-mediated increases in 4-HNE levels in the mouse placentas (Appendix A). 

To sum up, these data suggest that Elabela can effectively abolish the PE-like phenotype and partially abolish placental ferroptosis in the L-NAME-induced mouse model, suggesting that the underlying mechanism of Elabela may reverse PE by attenuating placental ferroptosis.

### 3.4. Erastin and RSL3 Induce HTR-8/SVneo Death in a Time-Dose-Dependent Manner

We first established a model of ferroptosis in HTR-8/Svneo cells. Previous literature on, e.g., hypoxia, H_2_O_2_, and LPS failed to induce the ferroptosis phenotype in HTR-8/Svneo cells in our study (Appendix A). Erastin and RSL3 are two classic inducers of ferroptosis [34], and we first demonstrated that Erastin and RSL3 could induce cell death in HTR-8/Svneo cells in a manner that is not inhibited by apoptosis inhibitors (Z-DEVD-FMK) and necrosis inhibitors (Necrostatin-1) but is rescued by Ferrostatin-1 and Deferoxamine-1 (Figure 4a and Appendix A). Moreover, the death of HTR-8/Svneo cells induced by Erastin and RSL3 was time- and dose-dependent (Figure 4b and Appendix A).

### 3.5. Elabela Can Rescue Ferroptosis Induced by Erastin

HTR-8/Svneo cells were treated with 5 μm of Erastin or 0.1 μM of RSL3, respectively, and different concentrations of Elabela were added. Surprisingly, we found that Elabela could only rescue Erastin-induced cell death (Figure 4c) but not RSL3-induced (Figure 2c), which may be related to the different mechanisms of action of the two inducers. We then found that Elabela could significantly save the cell viability reduced by short-term and low-concentration Erastin, and the rescue effect became weaker with the increase in the concentration and intervention time of Erastin (Figure 4d). After 24 h of treatment with high concentrations of Erastin (10 μM), Elabela could barely rescue the cells from death (Figure 4e).

### 3.6. Elabela Could Reduce Erastin-Exacerbated Oxidative Stress and Lipid Peroxidation, and Rescue Cell Dysfunction

We then investigated whether Elabela could rescue the typical ferroptosis phenotype of HTR-8/Svneo cells. Firstly, we found that Elabela reduced ROS (Figure 4f,g), MDA (Figure 4h) and Fe ion levels (Figure 4i,m), and increased GSH levels (Figure 4j) in trophoblast cells after Erastin treatment and reduced the degree of cellular lipid peroxidation (Figure 4h,l). Mitochondrial dysfunction was also a typical manifestation of ferroptosis. We found that the morphology of mitochondria was pyknotic and cristae were reduced or disappeared in Erastin-treated cells (Figure 4k), which could be effectively rescued by Elabela. Growing research has confirmed that trophoblast dysfunction leads to insufficient proliferation and invasion ability, then leads to shallow placenta implantation, which is among the pathological mechanisms leading to PE [3,4]. EdU staining revealed a downregulation of the proliferation capacity of treated cells relative to the control group (Appendix A), scratch assays revealed a decrease in cell migration capacity (Appendix A) and transwell assays revealed a decrease in cell invasion capacity (Appendix A).

### 3.7. Ferroptosis Is an Autophagy-Dependent form of Cell Death

To further evaluate the mechanism underlying the regulation of ferroptosis in placental trophoblasts by Elabela, HTR8/SVneo cells were treated with 5 μM Erastin (defined as control), 5 μM Erastin + 0.01 μM Elabela (defined as experimental) and transcriptome sequencing was performed. The sequencing results showed that there were 546 differential mRNAs in the experimental group compared with the control group (*p* < 0.01, FC > 2), including 120 mRNAs upregulated and 426 downregulated mRNAs (Figure 5a and Appendix A). The differential genes were further analyzed by KEGG and GO. KEGG analysis suggests that differential genes are associated with the regulation of many pathways, such as PI3K/AKT, autophagy, and ferroptosis (Appendix A). GO analysis showed that the differential gene biological functions were focused on blood pressure regulation and cell growth (Appendix A). Further GESA (*p* < 0.05 and qvalue < 0.25) analysis revealed that multiple pathways associated with autophagy were enriched (Figure 5b).

According to the sequencing analysis, we first investigated whether autophagy occurs in trophoblast cells under ferroptosis-inducing conditions. WB was performed after the treatment of trophoblast cells with Erastin. The results showed increased expression of Beclin1, ATG5, LC3B II, and P62 and decreased expression of NCOA4, the core proteins associated with the autophagic pathway, indicating activation and enhancement of autophagy (Figure 5d). 3-MA and CQ are two autophagy inhibitors with different mechanisms; 3-MA inhibits autophagy by inhibiting the formation of autophagosomes [35], while CQ blocks the patency of autophagic flow by inhibiting the fusion of autolysosomes with autophagosomes [36]. The results showed a decrease in Beclin1, ATG5 and LC3B II expression and an increase in NCOA4 expression after 3-MA treatment, which was consistent with CQ treatment, and an increase in LC3B II expression due to the inhibition of autophagosome fusion with lysosomes (Figure 5d). Ad-mCherry-GFP-LC3B is among the assays that can reflect the patency of autophagic flow. The green fluorescent protein (GFP), red fluorescent protein (mCherry), and a fusion protein of LC3B are efficiently expressed in the target cells after viral infection. When autophagy is enhanced, GFP fluorescence is quenched in the acidic environment of lysosomes during the process of fusion with autophagosomes. We found that Erastin significantly enhanced the process, while autophagy inhibition effectively rescued its intensity (Figure 5c). This indicates that autophagy can be induced under conditions of ferroptosis and that autophagic flow is unimpeded.

### 3.8. Ferritinophagy Is Involved in Ferroptosis in Trophoblasts

Mh Gao et al. previously found that under pathological conditions, NCOA4 attaches ferritin to autophagosomes, and then ferritin is degraded and releases its chelated iron; abundant LIP induces enhanced intracellular oxidative stress, which is called ferritinophagy [37,38]. To explore the specific mechanism, the expression of FTH1 in Erastin-treated cells was first examined. The result showed increased expression of FTH1, which contradicts our hypothesis (Figure 6a). Then, we exogenously transfected the FTH1-EGFP plasmid and detected the ectopic expression of FTH1 after Erastin treatment. The results suggested that Erastin reduced the expression of exogenous FTH1 (detected indicator GFP-FTH1) compared with the control group (Figure 6a), which was consistent with the fluorescence results (Figure 6d). Intervention with the autophagy inhibitor not only rescued the level of exogenous FTH1 but also increased the expression of endogenous FTH1 (Figure 6a). To further confirm the role of autophagy in iron death, we knocked down two key members of autophagy, ATG5 and NCOA4 [39].

After the sequence transfection efficiency was verified by WB (Figure 6b,f), we treated the transfected trophoblast cells with Erastin. Results revealed that knockdown of ATG5 and NCOA4 inhibited autophagy in trophoblast and also reversed the degradation of FTH1 compared to the control group (Figure 6c,g). Together with this, we also observed enhanced co-localization of LysoTracker (green) and FerroOrange (red) following Erastin stimulation, suggesting the presence of a large number of LIP in lysosomes (Figure 6e). Moreover, the inhibition of autophagy by either gene or drug could effectively reduce the amount of LIP and lysosomes (Figure 6e), indicating that ferritinophagy mediates the ferroptosis process.

### 3.9. The Ferritinophagy Pathway Plays a Role in the Regulation of Ferroptosis by Elabela

Elabela has been reported to rescue acute kidney injury in mice by reducing the levels of autophagy factors such as ATG5 and LC3B II [40]. We then further explored how Elabela regulates the mechanisms of autophagy and ferroptosis in trophoblasts. After Erastin treatment of HTR-8/Svneo cells, WB was performed and showed that TFR and FTH1 expression was upregulated and FPN1, SLC7A11, and GPX4 levels were decreased. After further treated with Elabela, the level of FTH1 was further increased (Appendix A). In addition, the degradation of exogenously expressed GFP-FTH1 was also inhibited (Figure 7a and Appendix A), which seems to have the same effect as inhibiting autophagy on FTH1 expression levels. The expression of autophagy-related genes reverted to varying degrees after the addition of Elabela compared to the treatment with Erastin alone (Figure 7a), and the number of LIP and lysosomes was reduced to a certain extent (Figure 7f), and the autophagic flow was also weakened (Figure 7g). Herein, we demonstrate that Elabela can rescue both autophagy and ferroptosis in trophoblast cells.

To clarify whether Elabela regulates ferroptosis through the ferritinophagy pathway, we constructed cell models in which ATG5 and NCOA4 were overexpressed. HTR-8/Svneo cells were treated with Erastin and Elabela, respectively, after WB verification of transfection efficiency (Figure 7b,d). The results showed that excessive activation of the autophagic pathway significantly impaired the rescue of exogenous FTH1 degradation by Elabela (Figure 7c,e). Meanwhile, the cell viability rescue by Elabela was also significantly reduced (Figure 7h). In brief, the ferritinophagy pathway plays a role in the regulation of ferroptosis by Elabela.

## 4. Discussion

PE is a placental-origin disease, and trophoblast dysfunction is among its key pathological mechanisms [41]. Ferroptosis is a new, adaptive and programmed cell death mode, which is characterized by the accumulation of a large amount of oxidized and toxic LIP, inducing the production of intracellular ROS, excessive oxidative stress leading to lipid peroxidation and consequent cell death [42]. The maternal requirement for iron increases significantly during pregnancy to support fetal growth and development. Studies have shown that excessive iron intake or high iron status is detrimental to pregnancy and is associated with pregnancies such as preeclampsia [11]. Therefore, whether ferroptosis in the placenta of PE is the crucial point we need to explore.

Previous studies have confirmed that the source of iron for lipid peroxidation during ferroptosis comes from the intracellular LIP, which induces a Fenton reaction that can produce highly toxic hydroxyl and peroxyl radicals [43]. In addition, MDA is the end product of intracellular lipid peroxidation and a marker of cellular oxidation status, which can indirectly reflect the level of intracellular oxidation and the intensity of ferroptosis [44]. Meanwhile, GSH is an important antioxidant in cells and an important regulator in scavenging intracellular free radicals and maintaining redox homeostasis [45]. They all reflect the degree of ferroptosis to a certain extent, and our results also suggest that these indicators are somewhat different in PE placentas compared to normal pregnant placentas. 4-HNE, an aldehyde product, is among the most representative metabolites of lipid peroxidation and can reflect the degree of lipid peroxidation. IHC staining suggests that 4-HNE is widely localized in CTBs, STBs and iEVTs in placental tissues and that 4-HNE expression is somewhat impaired in normal controls. These data suggest that the pathogenesis of PE is related to ferroptosis in the placenta.

Wang L and colleagues have found that both mRNA and protein levels of Elabela are reduced in the placentas of pregnancies with early-onset PE; therefore, Elabela has been considered a potential target for the prevention and treatment of this disorder [46]. In agreement with our results, PCR also suggested a decrease in Elabela transcript levels in PE placental tissue. Circulating levels of Elabela in pregnant women with PE were further examined and the results suggested that their levels were also significantly downregulated relative to normal. Considering that Elabela has a pathological mechanism that increases the proliferation and migration of vascular smooth muscle cells and affects the morphological changes in blood vessels and thus mediates the development of hypertension [47], it is reasonable to believe that this may also be among the pathological mechanisms of preeclampsia. IHC staining revealed that Elabela was mainly present in placental CTBs and STBs, and only minimally expressed in iEVTs, especially in placental iEVTs of preeclampsia, which is also consistent with the findings of Jing et al. [33]. Considering that syncytiotrophoblasts are the main sites of hormone secretion [48] and that Elabela acts as a peptide hormone, we, therefore, speculate that Elabela is mainly synthesized and secreted by syncytial trophoblasts, acting and influencing the function of extravillous trophoblasts in a paracrine manner. However, further experiments are needed to verify this. We, therefore, selected HTR8/Svneo to construct a cell model of ferroptosis to explore the link between Elabela and ferroptosis in PE.

Zhang Z et al. found that Elabela could alleviate the symptoms of hypertensive heart disease in mice by rescuing ferroptosis and reducing oxidative stress and promoting the proliferation and migration of cardiac vascular endothelial cells [27]. To investigate the function of Elabela in vivo, we constructed a PE-like model in mice using L-NAME [49], a nitric oxide synthase inhibitor. Treatment of pregnant mice in the second and third trimester with L-NAME mimicked similar pathological features of PE, such as hypertension, proteinuria, renal injury, and intrauterine growth restriction [50,51]. We, therefore, used this model to evaluate the effect of Elabela on PE in vivo. Przybyl et al. proposed that the labyrinth layer of the placenta is an important place to maintain the supply and exchange of nutrients for the fetus [52]. We measured the ratio of the labyrinth to junctional zone in mouse placentas. Compared with L-NAME-treated mice, Elabela effectively improved the reduction in the labyrinth to junctional layer ratio induced by L-NAME, which may be the mechanism by which Elabela ameliorates the reduction in fetal and placental weight. In addition, it has been reported that ferroptosis inhibitor Ferrostatin-1 can effectively alleviate ferroptosis levels to rescue the degree of PE phenotype in a reduced uterine perfusion pressure (RUPP) surgery constructed rat PE model [11]. Combined with our results, we hypothesized that Elabela also reversed the phenotypes of PE by reducing placental iron accumulation and lipotoxic damage in mice.

Including hypoxia [11], H_2_O_2_ [53], and LPS [54] are common conditions that induce a cell model of ferroptosis. Although they can increase the level of oxidative stress in trophoblast cells, they lack the essential features of ferroptosis in the present study. Erastin is the first identified ferroptosis activator, potent against human tumors containing oncogenes KRAS, HRAS, and BRAF mutations [55]. RSL3 is another molecule that was found to initiate ferroptosis by chemical screening [42]. First, we found that both agents could effectively reduce the viability of trophoblast cells, and this mode of cell death was distinct from apoptosis and necrosis. In addition, they reduced the viability of HTR8/Svneo cells in a time- and dose-dependent manner. Puzzlingly, Elabela was effective in rescuing the cell death induced by Erastin but barely rescued the induced by RSL3, which is consistent with the findings of Poli et al. [56]. This may be due to the fact that the targets of action of the two chemicals and the downstream pathways of cell death activation are inconsistent. Interestingly, Elabela could hardly rescue the cell death induced by the high concentration and longtime of Erastin. The level of iron ion, reactive oxygen species and lipid peroxidation were increased, and the function of the antioxidant system was decreased in Erastin-treated cell ferroptosis model. In addition, mitochondrial membrane density was increased and cristae decreased in trophoblasts. After adding Elabela intervention, we found that these indicators were rescued to a certain extent, consistent with previous studies [27], indicating that Elabela also has a corresponding effect on trophoblast ferroptosis.

Increasing evidence confirms ferroptosis can reduce the proliferation and invasion of tumor cells in numerous types of tumors [57,58,59]. Trophoblast cells have the characteristics of tumor-like behavior [60], and the results also suggest that ferroptosis leads to trophoblast dysfunction. At the same time, Elabela could partially rescue the phenotype of ferroptosis-induced cell dysfunction. These results suggest that the protective effect of Elabela against preeclampsia may be mediated by reducing ferroptosis in trophoblasts and improving cell function.

Usually, iron is present in the body as Fe^3+^, and its transport and storage are mainly regulated by the transferrin receptor (TFR) [61], ferritin (FTH) [62], and ferroportin (FPN) [63]. TFR is mainly responsible for the internalization of circulating Fe^3+^ into the cell, while FPN is mainly responsible for the externalization of intracellular Fe^3+^ into the cytoplasm. In addition, when there is an excess of iron ions in the cell, FTH can chelate with it, while on the contrary, the degradation of FTH promotes an increase in intracellular iron levels. The increased Fe^3+^ in cells readily undergoes single electron transfer reactions, which can be converted to Fe^2+^, which is highly oxidizing and toxic because it contributes to the production of hydroxyl radicals [5]. Gpx4 is a critical antioxidant enzyme that prevents the production of lipid hydrogen peroxide caused by elevated ROS [64]. Cystein-glutamate antiporter (System Xc−) is an important antioxidant system in cells, promoting cysteine uptake to synthesize GSH. Gpx4 takes GSH as a substrate to participate in the protective effect. When this system is inhibited, it reduces intracellular glutathione synthesis, leading to the accumulation of lipid hydrogen peroxide and subsequent cell death [65,66]. We found that the expressions of SLC7A11, GPX4 and FPN1 were downregulated and FTH1 and TFR were upregulated in Erastin-treated trophoblast cells. These results indicated that the Erastin-induced ferroptosis model of trophoblast cells might be induced by inhibiting XC-system to reduce the antioxidant effect while increasing the extracellular iron ion internalization and reducing the intracellular iron ion externalization to induce ferroptosis. FTH1 expression was increased in HRT8 cells treated with Erastin. After Elabela treatment, we found a further increase in FTH1 expression. Considering that FTH1 possesses iron chelation and therefore has the effect of reducing the LIP to rescue ferroptosis [67], we suggest that Elabela may rescue ferroptosis by increasing FTH1.

To further investigate the specific mechanism of Elabela in the regulation of ferroptosis, transcriptome profiling was performed. The results suggested that multiple autophagy pathways are involved in this regulation. Autophagy is a conserved physiological process for the turnover of intracellular substances during the evolution of eukaryotes. In this process, autophagosomes transport the proteins or organelles that need to be degraded to the autolysosome to complete degradation [68]. To explore the link between autophagy and ferroptosis, treatment of HTR8/Svneo cells with Erastin attenuated LC3B fluorescence spots and enhanced the co-localization of Fe^2+^ with lysosomes. At the same time, WB detection showed that the expression of core autophagy molecules, such as Beclin1, LC3B, and ATG5, was significantly upregulated, indicating that the typical autophagy mechanism also plays an important role in ferroptosis. Importantly, LIP was reduced by either genetic or pharmacological inhibition of autophagy. Intracellular iron homeostasis is a critical core of ferroptosis. It has been reported that intracellular iron homeostasis can be regulated by a specific NCOA4-mediated autophagy pathway called ferritinophagy. NCOA4 is utilized as a cargo receptor that recognizes ferritin and delivers it to lysosomes for degradation, leading to the release of free iron, which in turn exacerbates the LIP and mediates an increase in lipid peroxidation [38,69,70]. Paradoxically, FTH1 as a degradation substrate for ferritinophagy was increased in our study, contrary to the results of previous studies. Further ectopic expression of GFP-FTH1, followed by repeated experiments under the same conditions revealed that the ectopic expression of FTH1 was degraded, and this effect could be weakened by autophagy inhibition. Moreover, inhibition of autophagy can increase endogenous FTH expression, so the increase in FTH due to ferroptosis may be a negative feedback phenomenon mediated by the exacerbated LIP, which confirmed that Elabela rescued ferroptosis through an autophagy-dependent pathway. Furthermore, Elabela treatment effectively inhibited ferroptosis-induced autophagy and increased the expression of exogenous and endogenous FTH1. On the contrary, overexpression of autophagy-related genes revealed that Elabela could hardly rescue FTH1 degradation, and the rescue effect of Elabela on cell viability was abolished after excessive autophagy activation.

## 5. Conclusions

In summary, our results confirmed ferroptosis and Elabela as important pathophysiological factors involved in the development of PE. Elabela can disrupt ferritinophagy and inhibit NCOA4-mediated FTH1 degradation, thereby reducing the level of intracellular LIP and consequently reducing Fenton’s reaction and lipid peroxidation-mediated cellular dysfunction (Figure 8). This study provides us with new insights into the mechanism and therapeutic targets of PE.

## Figures and Tables

**Figure 1 cells-12-00099-f001:**
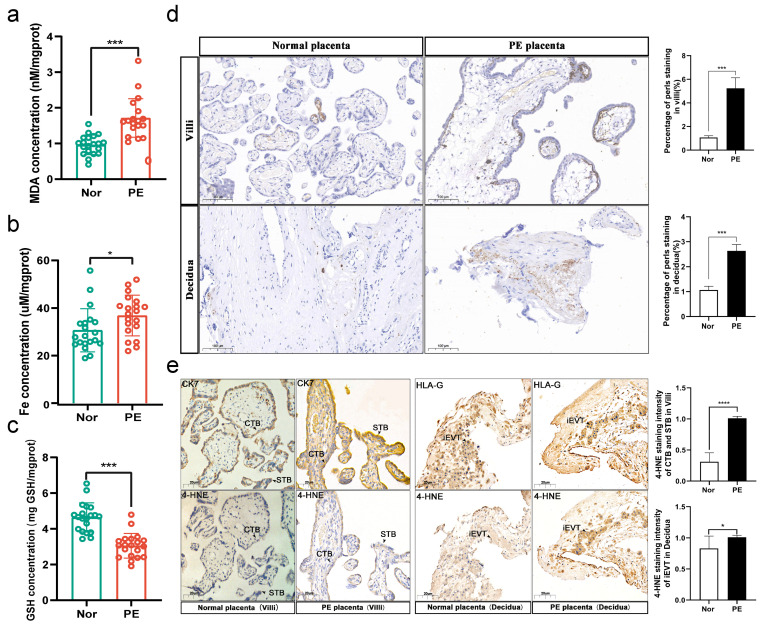
More severe ferroptosis presented in the placentas of PE. Placental tissues were collected from physiologically normal pregnancies (*n* = 20) and PE pregnancies (*n* = 20). (**a**) MDA concentration, (**b**) iron concentration, and (**c**) GSH concentration were measured using the corresponding detection kits, two-tailed *t*-test. (**d**) Perls’ Prussian blue staining for iron was performed in human PE and normal placentas. Quantification of staining intensity per patient; *n* = 3. Scale bars: 100 μm, two-tailed *t*-test. (**e**) IHC staining of 4-HNE in human PE and normal placentas. Quantification of staining intensity per patient; *n* = 3. Scale bars: 20 μm, two-tailed *t*-test. iEVTs and CTBs were identified by HLA-G and CK7 staining, respectively. STB, syncytiotrophoblasts; iEVT, interstitial extravillous trophoblast; CTBs, cytotrophoblasts; CK7, cytokeratin 7; HLA-G, human leukocyte antigen G. All data are presented as the means ± SEM. * *p* < 0.05; *** *p* < 0.001; **** *p* < 0.0001.

**Figure 2 cells-12-00099-f002:**
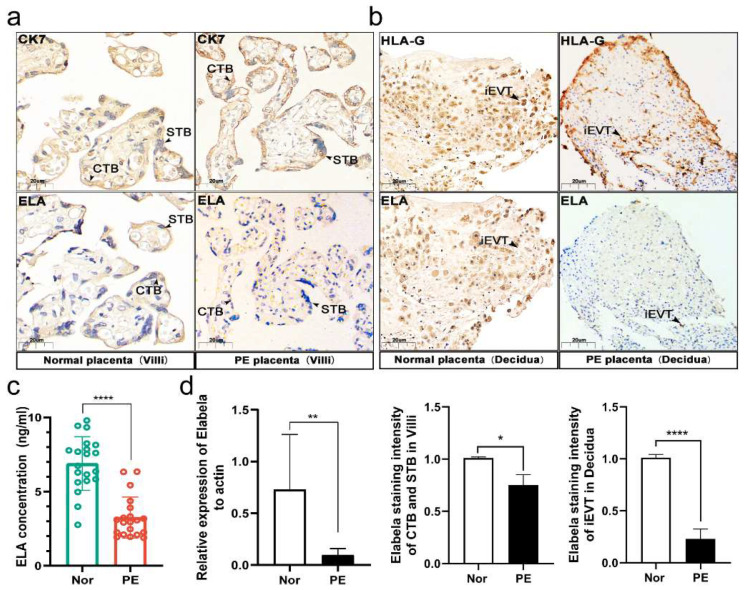
Low levels of Elabela in the placenta and circulation of the pregnancy with preeclampsia. Placental tissues and blood were collected from physiologically normal pregnancies (*n* = 20) and PE pregnancies (*n* = 20). (**a**,**b**) IHC staining of Elabela in human PE and normal placentas. Quantification of staining intensity per patient; *n* = 3. Scale bars: 20 μm, two-tailed *t*-test. (**c**) ELISA analysis of the expression of Elabela in the circulation of PE and normal pregnancies, two-tailed *t*-test. (**d**) RT-qPCR analysis of the expression of Elabela in preeclampsia and normal placentas, two-tailed *t*-test. ELA, Elabela. All data are presented as the means ± SEM. * *p* < 0.05; ** *p* < 0.01; **** *p* < 0.0001.

**Figure 3 cells-12-00099-f003:**
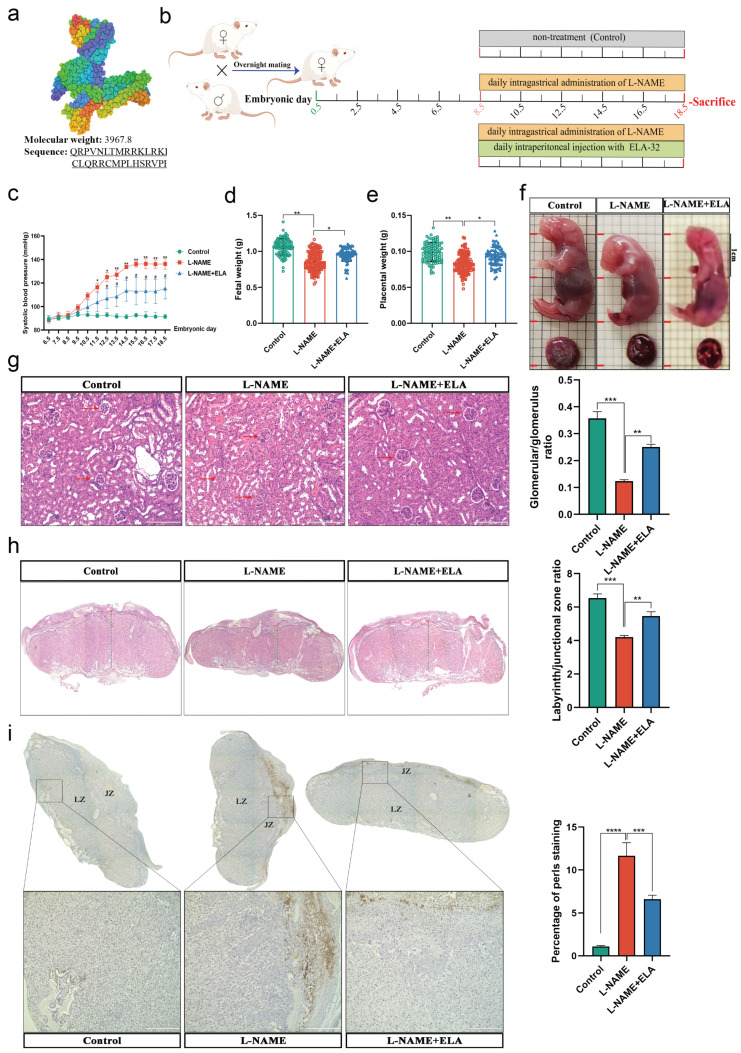
The PE-like and ferroptosis phenotypes in mice were relieved by Elabela administration. (**a**) The biological characteristics of Elabela. (**b**) Schematic illustration of the experimental design. (**c**) Systolic blood pressure of pregnant mice in control (*n* = 7), L-NAME(*n* = 10), and L-NAME+ELA group (*n* = 7), one-way ANOVA and Tukey’s multiple comparison test. (**d**) Fetal weight; *n* = 94 fetuses from 7 dams in the control group, *n* = 134 fetuses from 10 dams in the L-NAME group, and *n* = 81 fetuses from 7 dams in the L-NAME+ELA group, one-way ANOVA and Tukey’s multiple comparison test. (**e**) Placental weight; *n* = 94 fetuses from 7 dams in the control group, *n* = 134 fetuses from 10 dams in the L-NAME group, and *n* = 81 fetuses from 7 dams in the L-NAME+ELA group, one-way ANOVA and Tukey’s multiple comparison test. (**f**) Representative images of the fetuses of the control, L-NAME, and L-NAME+ELA groups; scale bars:1 cm. (**g**) H&E staining of maternal kidney on E18.5, and measurement of Bowman space; *n* = 3. Scale bars: 200 μm, one-way ANOVA and Tukey’s multiple comparison test. (**h**) H&E staining of placental sections at E18.5. The LZ and JZ areas and the Lz/Jz ratio were quantified; *n* = 3. Scale bars:2000 μm, one-way ANOVA and Tukey’s multiple comparison test. (**i**) Perls’ Prussian blue staining of placental sections at E18.5, Quantification of staining intensity per placentas; *n* = 3. Scale bars: 2000 μm for the up panel and 400 μm for the bottom panel, one-way ANOVA and Tukey’s multiple comparison test. LZ, Labyrinth zone; JZ, Junction zone. All data are presented as the means ± SEM. * *p* < 0.05; ** *p* < 0.01; *** *p* < 0.001; **** *p* < 0.0001.

**Figure 4 cells-12-00099-f004:**
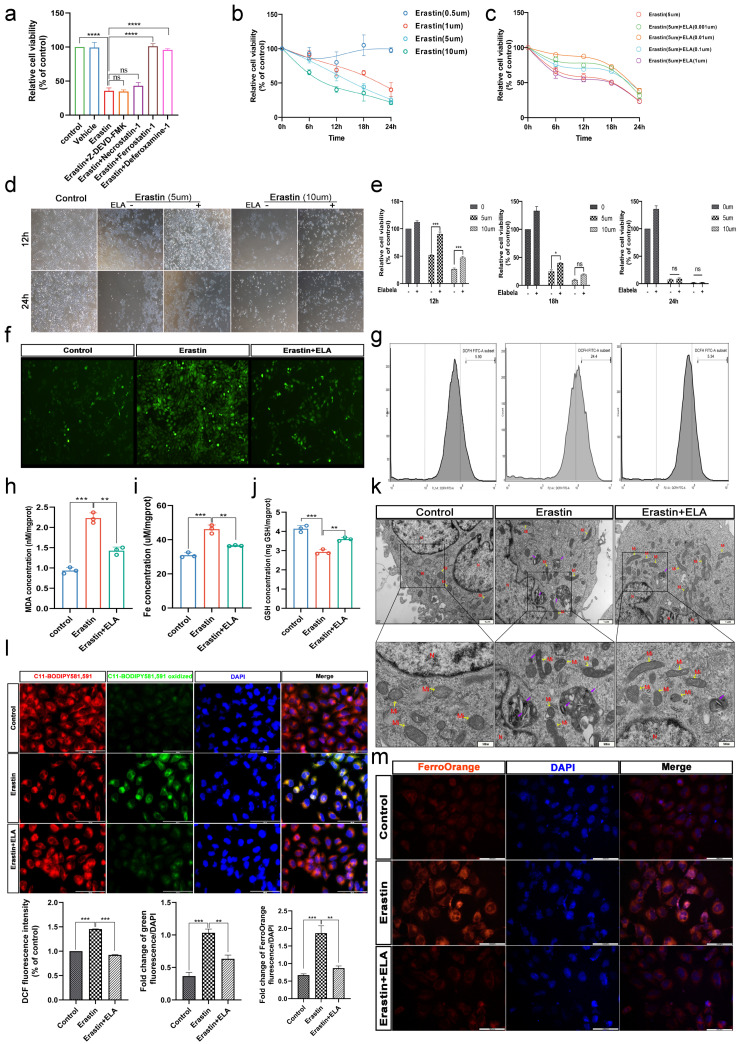
Elabela can rescue ferroptosis induced by Erastin. Construction of the ferroptosis models using Erastin treatment of HTR-8/Svneo cells. (**a**) Pretreatment with apoptosis inhibitor (Z-DEVD-FMK, 20 μM), necrosis inhibitor (Necrostatin-1, 0.5 μM), and ferroptosis inhibitor (Ferrostatin-1, 60 nM; Deferoxamine-1 100 μM) for 1 h, Erastin (5 μM) was added into cells, and 18 h later, cell viability was determined by CCK8; *n* = 6. One-way ANOVA and Tukey’s multiple comparison test. (**b**) After treatment with different concentrations of Erastin for different times, CCK8 was used to determine cell viability; *n* = 6. one-way ANOVA and Tukey’s multiple comparison test. (**c**) Pretreatment with different concentrations of Elabela for 1 h, then Erastin (5 μM) were added into HTR-8/Svneo cells for treated different times, and cell viability was measured by CCK8; *n* = 6. One-way ANOVA and Tukey’s multiple comparison test. (**d**) Representative images showing the induction of cell death treated in Erastin (5 μM, 10 μM) with or without Elabela (0.01 μM) for 12/24 h. Scale bars, 400 μm. (**e**) HTR-8/Svneo cells were treated with Erastin (5 μM, 10 μM) with or without Elabela (0.01 μM) for 12 to 24 h. Cell viability was detected by CCK8 assay; *n* = 6. Two-tailed *t*-test. (**f**–**m**) After pretreatment with Elabela (0.01 μM) for 1 h, the cells were added Erastin (5 μM) for treated 12 h. (**f**) Measuring the fluorescence intensity of the DCFH-DA probe among groups by microscopy; *n* = 3. (**g**) Flow cytometry assay for measuring ROS by staining with DCFH-DA, normalized by the number of cells uploaded, *n* = 3. One-way ANOVA and Tukey’s multiple comparison test. (**h**) MDA concentration, (**i**) iron concentration, and (**j**) GSH concentration were measured using the corresponding detection kits; *n* = 3. One-way ANOVA and Tukey’s multiple comparison test. (**k**) Transmission electron microscopy images of organelles. Scale bars: 1 μm for the up panel and 500 nm for the bottom panel; *n* = 3. (**l**) Measuring cellular lipid peroxidation by fluorescence microscopy using the C11 BODIPY 581/591 fluorescent probe. Total C11 BODIPY 581/591 (red), oxidized C11 BODIPY 581/591(green), DAPI (blue) stained nucleus, *n* = 3. Scale bar: 100 μm. One-way ANOVA and Tukey’s multiple comparison test. Intracellular Fe^2+^ was measured by fluorescence microscopy using the FerroOrange fluorescent probe. FerroOrange (red) stains Fe^2+^ and DAPI (blue) stains the nucleus; *n* = 3. Scale bar: 50 μm. One-way ANOVA and Tukey’s multiple comparison test. *n*, nucleus; Mi, mitochondria. All data are presented as the means ± SEM. ns, non-significant; * *p* < 0.05; ** *p* < 0.01; *** *p* < 0.001; **** *p* < 0.0001.

**Figure 5 cells-12-00099-f005:**
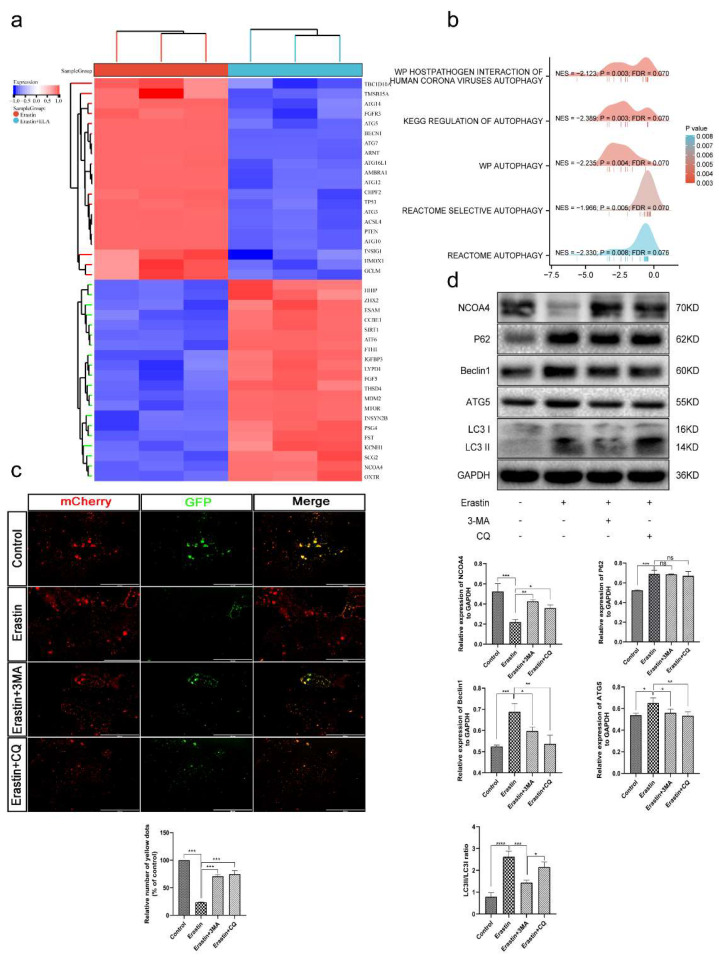
Ferroptosis is an autophagy-dependent form of cell death. (**a**) Heatmap of the significant differences expressed genes between the experimental group (Erastin+Elabela) and the control group (Erastin). (**b**) GSEA analysis of genes. (**c**) HTR-8/Svneo cells was transfected with Ad-mCherry-GFP-LC3B adenovirus (20 MOI) for 24 h. Next, Cells were pretreated with 3-MA(2 mM), CQ (50 μm) for 1 h, respectively, and Erastin (5 μm) was added for 12 h. Fluorescence images of GFP/mCherry spots were examined; *n* = 3. Scale bars: 400 μm. One-way ANOVA and Tukey’s multiple comparison test. (**d**) HTR-8/Svneo cells were pretreated with 3-MA (2 mM), CQ (50 μm) for 1 h, respectively, and Erastin (5 μm) was added for 12 h, and protein levels of NCOA4, P62, Beclin1, ATG5, and LC3were examined by Western blotting; *n* = 3, one-way ANOVA and Tukey’s multiple comparison test. All data are presented as the means ± SEM. ns, non-significant; * *p* < 0.05; ** *p* < 0.01; *** *p* < 0.001; **** *p* < 0.0001.

**Figure 6 cells-12-00099-f006:**
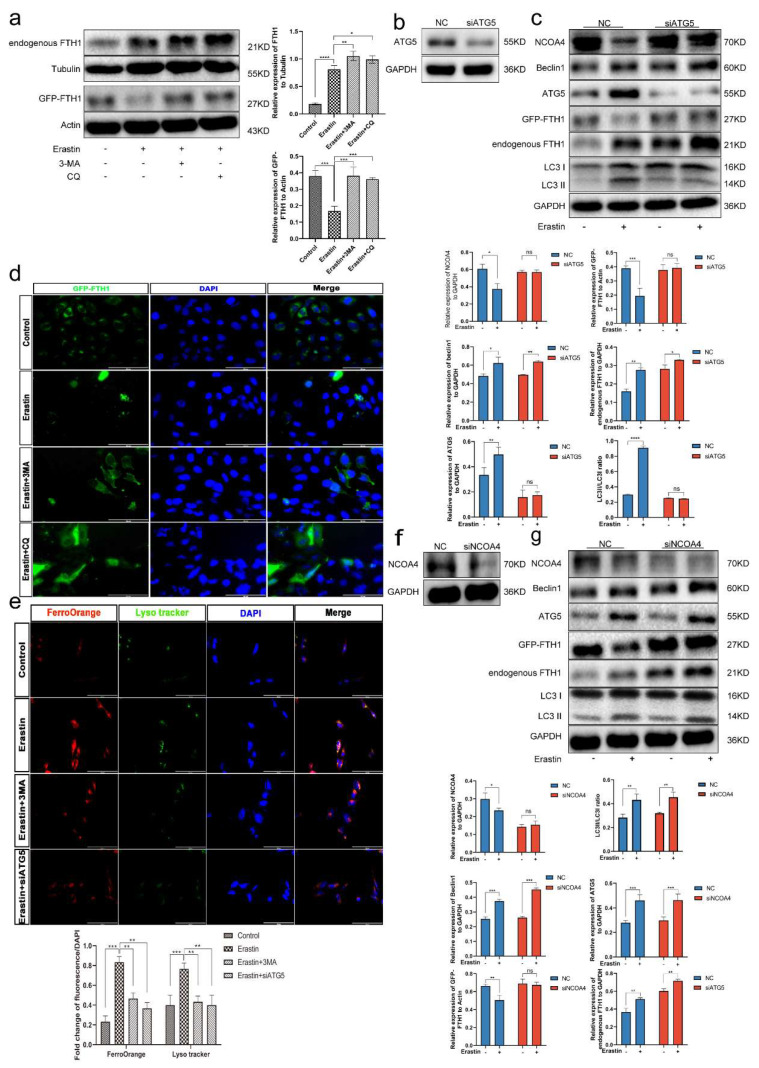
Ferritinophagy is involved in ferroptosis in trophoblasts. (**a**) HTR-8/Svneo cells were transfected with FTH1-EGFP plasmid for 24 h. Next, Cells were pretreated with 3-MA(2 mM), CQ (50 μm) for 1 h, respectively, and Erastin (5 μm) was added for 12 h, and protein levels of endogenous FTH1 and exogenous FTH1 were examined by Western blotting; *n* = 3. (**b**) WB validates si-AGT5 interference efficiency. (**c**) HTR-8/Svneo cells were interfered with si-ATG5 and then treated with Erastin (5 μm) for 12 h. WB detection of the protein levels of NCOA4, Beclin1, ATG5, LC3B, endogenous FTH1, and exogenous FTH1. (**d**) Fluorescence images of exogenous FTH1 (green) after treatment as indicated. DAPI (blue) stained nucleus, Scale bars: 100 μm. (**e**) After treatment, fluorescence images of lysosomes (green) and Fe^2+^ (red) co-localization in HTR-8/Svneo cells as indicated; *n* = 3. Scale bars: 100 μm. (**f**) WB validates si-NCOA4 interference efficiency. (**g**) HTR-8/Svneo cells were interfered with si-NCOA4 and then treated with Erastin (5 μm) for 12 h. WB detection the protein levels of NCOA4, Beclin1, ATG5, LC3B, endogenous FTH1 and exogenous FTH1; *n* = 3. one-way ANOVA and Tukey’s multiple comparison test. All data are presented as the means ± SEM. ns, non-significant; * *p* < 0.05; ** *p* < 0.01; *** *p* < 0.001; **** *p* < 0.0001.

**Figure 7 cells-12-00099-f007:**
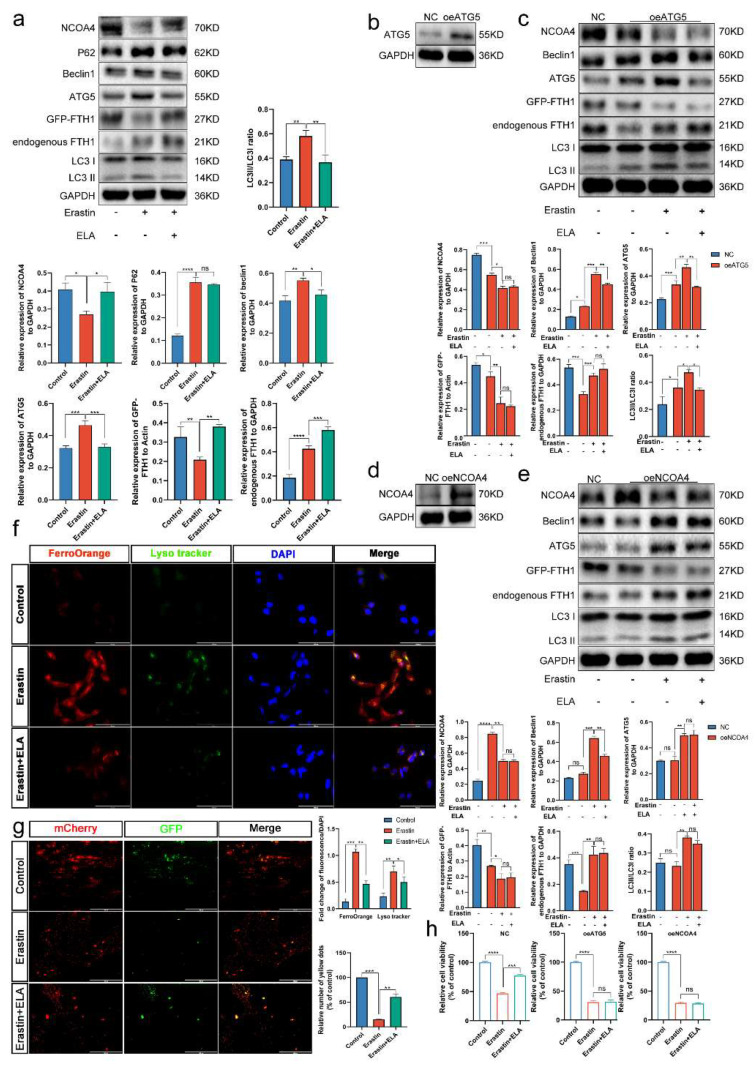
The ferritinophagy pathway plays a role in the regulation of ferroptosis by Elabela. (**a**) HTR-8/Svneo cells treatment with Erastin (5 μm) and Elabela (0.01 μm) for 12 h. WB detection the protein levels of NCOA4, Beclin1, ATG5, LC3B, P62, endogenous FTH1 and exogenous FTH1; *n* = 3. (**b**) WB validates OE-ATG5 plasmid interference efficiency. (**c**) HTR-8/Svneo cells were interfered with by OE-ATG5 plasmid and then treated as indicated, wb detection of the protein levels of NCOA4, Beclin1, ATG5, LC3B, endogenous FTH1, and exogenous FTH1; *n* = 3. (**d**) WB validates OE-NCOA4 plasmid interference efficiency. (**e**) HTR-8/Svneo cells were interfered with by OE-NC0A4 plasmid and then treated as indicated, wb detection of the protein levels of NCOA4, Beclin1, ATG5, LC3B, endogenous FTH1, and exogenous FTH1; *n* = 3. (**f**) After treatment, fluorescence images of lysosomes(green) and Fe2+(red) co-localization in HTR-8/Svneo cells as indicated; *n* = 3. Scale bars: 100 μm. (**g**) After treatment, fluorescence images of GFP/mCherry spots as indicated; *n* = 3. Scale bars: 400 μm. (**h**) After overexpression of ATG5 and NCOA4, the cells were treated with Erastin (5 μm) and Elabela (0.01 μm) for 12 h, and CCK8 was used to measure cell viability; *n* = 6. One-way ANOVA and Tukey’s multiple comparison test. All data are presented as the means ± SEM. ns, non-significant; * *p* < 0.05; ** *p* < 0.01; *** *p* < 0.001; **** *p* < 0.0001.

**Figure 8 cells-12-00099-f008:**
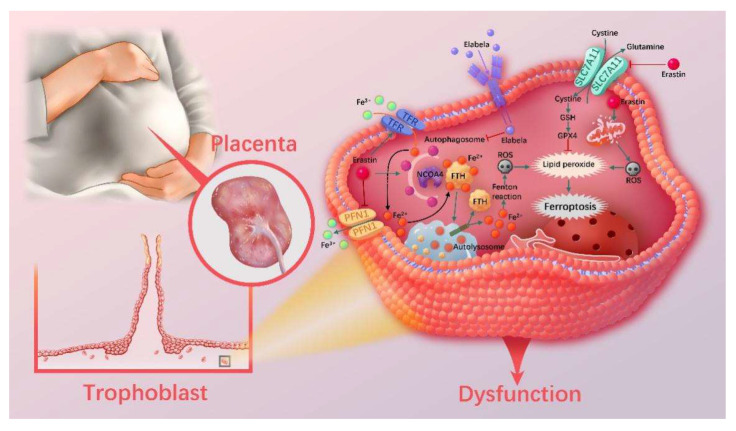
Schematic illustration of the mechanism of how Elabela and Erastin regulate ferroptosis in trophoblasts.

## Data Availability

Informed written consent was obtained from all patients. All data generated from this study including supporting information and Raw data are available from the corresponding author on reasonable request.

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
