# Peer review of "Elabela: Negative Regulation of Ferroptosis in Trophoblasts via the Ferritinophagy Pathway Implicated in the Pathogenesis of Preeclampsia"

_cells, 2022, doi:10.3390/cells12010099_

Round 1

Reviewer 1 Report

The article is well designed, presented and described. I only suggest some minor revision about the definition of PE.

The introduction provides sufficient background. 

Lines 78-85 this part should be placed in “Conclusions” section and not in the introduction

The introduction should end with your aims.

PE presents heterogeneity in clinical presentation, disease severity, placental pathology, and pathophysiological mechanisms. Could you specify the clinical features of patients enrolled in this study? These informations could be useful to understand which kind of PE has been analyzed. 

What is the proportions of Fetal growth restriction? of early an late-PE?

Did you perform a subgroup analysis in PE group, according to different features?

Materials and Results are widely described. Well done

Line 570-573 this part should be deleted

Reviewer 2 Report

The goal of the paper is the role of Elabela - novel polypeptide and it’s role in preeclampsia. The study clearly shows that deficiency of Elabela exacerbates ferroptosis in the placenta which might be one of potential mechanisms in the pathogenesis of preeclampsia. Also editorial part of the paper was of high quality and all tables, photos and graphs were very readable. Methodology was proper and only one thing could be improved - enlarge the group of studied patients (n=20). The references were new - mostly after 2015 and appropriate.
